**Data Availability Statement:** Data cannot be shared publicly because we do not have ethics

---

# "I think we did the best that we could in the space:" A qualitative study exploring individuals' experiences with three unconventional environments for patients with a delayed hospital discharge

Michelle Marcinow[1‡], Lauren Cadel[1,2‡*], Arija Birze[1], Jane Sandercock[1,3], Junhee Baek[1], Walter Wodchis[1,4], Sara J. T. Guilcher[2,4‡], Kerry Kuluski[1,4‡]

1 Institute for Better Health, Trillium Health Partners, Mississauga, Ontario, Canada, 2 Leslie Dan Faculty of Pharmacy, University of Toronto, Toronto, Ontario, Canada, 3 Faculty of Rehabilitation Science, McMaster University, Hamilton, Ontario, Canada, 4 Institute of Health Policy, Management and Evaluation, University of Toronto, Toronto, Ontario Canada

‡ MM & LC are co-first authors on this work. SJTG & KK are co-senior authors on this work.

* lauren.cadel@utoronto.ca

## Abstract

### Background

Given growing hospital capacity pressures, persistent delayed discharges, and ongoing efforts to improve patient flow, the use of unconventional environments (newly created or repurposed areas for patient care) is becoming increasingly common. Despite this, little is known about individuals' experiences in providing or receiving care in these environments.

### Objectives

The objectives of this study were to: (1) describe the characteristics of three unconventional environments used to care for patients experiencing a delayed discharge, and (2) explore individuals' experiences with the three unconventional environments.

### Methods

This was a multi-method qualitative study of three unconventional environments in Ontario, Canada. Data were collected through semi-structured interviews and observations. Participants included patients, caregivers, healthcare providers, and clinical managers who had experience with delayed discharges. In-person observations of two environments were conducted. Interviews were transcribed and notes from the observations were recorded. Data were coded and analyzed thematically.

### Results

Twenty-nine individuals participated. Three themes were identified for unconventional environments: (1) implications on the physical safety of patients; (2) implications on staffing models and continuity of care; and, (3) implications on team interactions and patient care.

approval or participant consent to share data publicly. Please contact the Trillium Health Partners Research Ethics Board at THPREB@thp.ca to request access to the de-identified data for researchers who meet the criteria for access to confidential data.

**Funding:** This research was funded by the Canadian Institutes of Health Research Operating Grant on transitions in care (165593). The funders had no role in study design, data collection and analysis, decision to publish, or preparation of the manuscript.

**Competing interests:** The authors have declared that no competing interests exist.

Participants discussed how the physical set-up of some unconventional spaces was not conducive to patient needs, especially those with cognitive impairment. Limited space made it difficult to maintain privacy and develop social relationships. However, the close proximity of team members allowed for more focused collaborations regarding patient care and contributed to staff fulfilment. A smaller, consistent care team and access to onsite physicians seemed to foster improved continuity of care.

## Conclusions

There is potential to learn from multi-stakeholder perspectives in unconventional environments to improve experiences and optimize patient care. Key considerations include keeping hallways and patient rooms clear, having communal spaces for activities and socialization, co-locating team members to improve interactions and access to resources, and ensuring a consistent care team.

## Introduction

Patients experiencing a delayed care transition, also known as a delayed discharge or alternate level of care, are a population that was frequently transitioned to unconventional environments during the pandemic [1]. Unconventional environments are a model of care and can be conceptualized as spaces that are repurposed with beds or stretchers to provide patient care (e.g., hallways, offices, visitor lounges, conference rooms) [2]. These spaces often do not meet safety standards and lack sufficient resources (e.g., nurse call-bells, staffing levels) [2]. Patients with delayed discharges often do not have acute medical needs, but remain in hospital while waiting for their next point of care, such as a long-term care bed or home with services [3]. These patients are medically ready to be discharged from hospital, but do not have a safe place to transition to. Discharge delays are a quality of care issue, with consequential impacts to hospital admissions, capacity issues in emergency rooms, and delayed or cancelled surgeries and treatments [4–6]. Unfortunately, while patients remain in hospital they often experience poor outcomes, including functional and cognitive decline and suboptimal experiences [7–10].

For many years, hospitals in Canada and globally have been struggling with capacity pressures (e.g., admission surges due to overcrowded emergency departments and staffing shortages), which were further exacerbated by the COVID-19 pandemic [11–14]. In anticipation of inpatient surges from COVID-19, hospitals across Canada aimed to reduce in-patient occupancy during the earlier stages of the pandemic and it was reported that some Canadian hospitals (which typically operate close to or over 100% capacity) reduced their occupancy to 50% between March 2020 and June 2020 [15–17]. While some hospitals lowered occupancy by rapidly discharging patients, others added more beds in intensive care units [18], or through the use of unconventional environments (hallways, offices, auditoriums, gymnasiums, tents, hotels, etc.) [17, 19]. Patients with delayed discharge may not have to meet a single set of criteria to be admitted or transferred to an unconventional environment but required to meet eligibility criteria (e.g., ambulatory, no cognitive impairment) set by the location (e.g., hospital, unit).

While outcomes and experiences have been explored in conventional hospital environments, it is not known how individuals, such as patients, caregivers, and healthcare providers experience care in unconventional environments. While, there has been some research to explore the use of unconventional spaces for other purposes (e.g., hotel or exhibition centre

repurposed to provide care for COVID-19 patients) [20–22], to our knowledge, there are no studies exploring the use of these environments for patients experiencing delayed discharge during the COVID-19 pandemic. Given continued hospital capacity pressures, it is likely that unconventional spaces will continue to exist within the care continuum. It is important to understand what works well and what can be improved upon to ensure patients receive quality care going forward. Therefore, the objectives of this study were to: (1) describe the characteristics of three unconventional environments (decommissioned hospital, hotel, tent) and (2) explore individuals' experiences with the three unconventional environments.

## Methods

### Study design

Our research team conducted an exploratory, multi-method qualitative descriptive study [23] to understand how individuals experienced providing, or receiving, care in unconventional environments for patients who had a delay in hospital discharge. Our multi-methods approach included interviews and on-site observations that focused on understanding individuals' experiences in receiving and delivering care in an unconventional hospital environment, specifically observing processes related to patient care, discharge planning, and care transitions. We interviewed participants from three different unconventional environments that were associated with two main hospital networks: Site A, a decommissioned hospital that had been unoccupied and re-opened to help relieve capacity issues (physically away from, but affiliated with, the main hospital); Site B, a hotel previously used for cancer patient accommodations while they were receiving care away from home (located beside the main acute hospital site); and Site C, a temporary structured, heated tent (located in a hospital parking lot). We also conducted in-person unit observations in Site A and B once provincial COVID-19 restrictions were lifted to collect detailed descriptions of the sites. No observations were conducted at Site C because researchers were not allowed to enter the site due to COVID-19 related restrictions. The tent was a temporary structure set up to help alleviate hospital capacity and flow pressures during a time when strict provincial COVID-19 restrictions were implemented (no patients with COVID-19 were cared for in this setting). Our study was approved by the Research Ethics Boards of the two hospital networks where data collection took place and the University of Toronto. All participants provided written or verbal informed consent prior to participating and individuals at the observation sites were informed of the researchers' presence.

### Participants and recruitment

*Unit level observations*: A member of the research team contacted unit managers of each study site by phone or videoconferencing (i.e., Zoom) to explain the purpose of the study. Our team did not interfere with daily operations during the pandemic. Initial contact was followed by an e-mail for logistical planning and to confirm details.

*Interviews*: Participants interviewed included patients with a delayed discharge, caregivers, care providers (e.g., physicians, nurses, rehabilitation therapists), and staff who held a clinical manager role who had experience with delayed discharges and transition planning in the three unconventional environments. All participants had to be 18 years of age or older, English speaking, and able to provide consent. Participants were recruited from three unconventional environments associated with a main acute hospital network. All sites were located in Ontario, Canada (both northern and southern parts of the province). Purposive and convenience sampling strategies were used to recruit participants through the research teams' professional contacts at each main hospital network. Snowball sampling was used to identify other individuals interested in participating. A member of the research team contacted potential participants by

telephone to see if they would be interested in participating. Participants were also recruited during unit observations. Participants had no prior relationship with the interviewers but were informed of the reasons for conducting the study prior to participation. No participants withdrew from the study and individuals who chose not to participate were not tracked.

## Data collection

*Unit level observations*: Two members of the research team (AB, HS) conducted observations at Site A and B to observe processes as they related to patient care, discharge planning, and care transitions in unconventional environments between May 2022 and August 2022. Observations were made during rounds, meetings, huddles, and general activities on the unit. Observations and observational field notes followed a semi-structured guide that was developed to systematically record data related to providing and receiving care in the unconventional environments, with unlimited opportunity for unstructured, discovery-oriented observations. Observations lasted from one to eight hours per visit, totalling 20 hours and field notes were taken by team members to provide rich detailed descriptions of events, interactions, physical environment, and use of space on the unit. Field notes collected did not include any names or identifying information about hospital staff, patients, or family members. Team members obtained written consent from the unit manager requesting approval for field observations and worked closely with the manager to ensure that unit staff were notified and comfortable with the visit. An information card with written information about the purpose of the field observations was provided to unit staff and verbal consent was obtained from hospital staff prior to field observations. No staff members declined consent for observations.

*Interviews*: Research team members (JS, HS, LC, MM) conducted in-depth, semi-structured interviews with all participants between June 2021 and August 2022. All interviewers were trained in conducting qualitative interviews and received supervision and mentorship from the Principal Investigators (KK, SJTG). Participants provided verbal informed consent before participating in an interview. Interviews were conducted by telephone or videoconferencing (i.e., Zoom) and lasted between 25 and 70 minutes in length. For interviews conducted by videoconference, the interviewer had their camera on, but participants had the option to be on camera based on their preferences. Interview guides were developed to explore participant experiences with receiving or providing care in an unconventional environment (one for providers and clinical managers and one for patients and caregivers). More specifically, the following topics were explored in the interviews: day-to-day work activities, team dynamics and communication, strategies to address delayed discharge, gaps in care, experience of care received in hospital, and recommendations for improvement. To encourage sharing of individual experiences, recruited participants were not paired (e.g., providers were not paired with their clinical manager and patients were not paired with their caregiver). The interviewers made reflexive notes following each interview and discussed each transcript during weekly team meetings to inform subsequent interviews. All interviews were audio-recorded and transcribed for analysis. During transcription, all participants were assigned a pseudonym.

## Data analysis

*Interviews*: Data collection and analysis were conducted concurrently until saturation was achieved, which occurred when no new major concepts were discussed [24]. A thematic analysis [25] was conducted to identify common themes across the interviews related to participants' experiences and perceptions of the three unconventional environments. Interviews were analyzed inductively, where interviewer statements were used to capture new concepts, as well as deductively by identifying challenges and facilitators related to receiving or providing

care in the unique environments. Deductive coding was informed by a leading practices guide (updated 2021) for delayed discharge (*Alternate Level of Care Leading Practices Guide*: *Preventing Hospitalization and Extended Stay for Older Adults)* [26, 27], with codes aligning with its components. This guide was developed by a working group, comprised of healthcare providers, directors, managers, quality leads, researchers, and advisors, and uses evidence-based leading practices to inform the proactive management and care provided to hospitalized older adults to prevent delays in care transitions to post-acute setting or back home. It contains a list of leading practices, along with key tools and resources that align with three key goals: integrating senior friendly care across organizations, ensuring practices are in place in the emergency department to prevent unnecessary hospital admissions, and avoiding hospital-acquired harm. A subset of transcripts was divided amongst the research team (MM, LC, JS, KK, SJTG) to identify key concepts from the interviews. These concepts were used to inform a preliminary codebook that was used by three team members (MM, LC, JS) to code a subset of transcripts before finalizing and applying the codebook to the remaining transcripts. For the purposes of the study objectives the following codes were further analyzed by team members (MM, LC, AB): Perceptions of the model (i.e., care provided in the unconventional environment); opportunities for improvement; and, staff or patient fulfillment/satisfaction. Through an iterative process, the research team identified overarching concepts that evolved through further reflection in a series of team meetings to form the themes discussed in this paper.

*Unit level observations*: In combination with publicly available documents (e.g., pamphlets, brochures, information sheets, press releases), observational field notes from sites were used to provide an understanding of physical space, use of space on the units, events, and interactions. Subsequent to interview coding and team discussions, field notes were coded paying specific attention to how observational data could inform our understanding of the codes of interest identified in the interview data. Additional codes were used to inform the descriptions outlined in the results: context of the model of care, physical space, staff composition, patient eligibility, and participant role. Credibility is a criterion for trustworthiness and was established through data triangulation of different sources (interviews, observations, documents) [28]. Data were compared in a table in Microsoft Excel for similarities and differences.

## Results

A total of 29 individuals participated in this study. Across all three unconventional environments, we interviewed six care providers (e.g., nurses, physicians), eight allied health professionals (e.g., social workers, rehabilitation therapists, physical and occupational therapists), and six staff in a clinical manager role (e.g., charge nurse, manager). We interviewed seven caregivers and two patients from two of the care environments (Site A and Site B), as we were unable to recruit patients or caregivers who received care in the tent structure (Site C) due to COVID-19 restrictions. Table 1 provides a summary of characteristics of each unconventional environment, including historical context, physical environment, proximity to main hospital, patient eligibility, and staff composition.

Three main themes were identified to capture experiences with the three unconventional environments for patients with delayed discharge: (1) Implications on the physical safety of patients; (2) Implications on staffing models and continuity of care; and (3) Implications on team interactions and patient care (see Table 2).

### Implications on the physical safety of patients

Participants highlighted aspects of the unconventional environments that created perceived unsafe situations for both patients and caregivers. Safety concerns were discussed in the

**Table 1. Characteristics of the three unique environments[i].**

| Site | Context | Characteristics | | | |
|---|---|---|---|---|---|
| | | Physical Environment[i] | Location/ Proximity to Main Hospital Site | Patient Eligibility | Staff Composition |
| Site A | Decommissioned hospital, reopened to address capacity pressures | • Floors: 3 floors, 1 locked unit<br>• Rooms: Semi-private and private; crowded<br>• Beds: 99 beds | Building not adjacent to main hospital (30 minutes+ by car; 90 minutes by public transit) | • All patients designated with delayed discharge<br>• Can have cognitive impairment and responsive behaviours | Interdisciplinary team without geriatric specialization |
| Site B | Hotel used for out-of-town cancer patients receiving treatment. Repurposed to address capacity pressures | • Floors: 1 floor• Rooms: Mostly semi-private rooms; small<br>• Beds: 20 beds | Building is adjacent to main hospital | • Elderly patients with restorative potential[ii]<br>• Experience 1+ geriatric syndromes[iii]<br>• Ambulatory | Interdisciplinary team with geriatric specialization |
| Site C | Tent; originally opened at the start of the COVID-19 pandemic to prepare for capacity challenges | • Temporary structure• Floors: 1 floor• Rooms: Not applicable<br>• Beds: 50 beds | Parking lot adjacent to main hospital, connected with covered tunnel | • Patients with restorative potential<br>• Ambulatory | Interdisciplinary team without geriatric specialization |

[I] Characteristics of the environments were drawn from interview and observational data

[ii] Patients ineligible if: they require oxygen therapy, suction, telemetry, wheelchair for primary mobility, cannot descend one flight of stairs, are over 300lbs, are waiting for long-term care, have a wander risk, are hospitalized for mental health

[iii] Geriatric syndromes: Cognitive impairment, polypharmacy, caregiver stress, impaired mobility

interviews and also noted in the observations in terms of physical hazards that could lead to injury risks for patients, gaps in essential care infrastructure, and inadequate space to provide comprehensive patient care. Some of the environments were observed to be small and lacked sufficient storage space. Equipment and assistive devices were left in hallways or patient rooms, crowding hallways and blocking exits and patient beds. For example, a healthcare provider from Site A explained how fitting patient beds and necessary equipment in the rooms was challenging based on their small size and physical layout, which impacted how care was provided:

*I mean the hospital itself was built [over 60 years ago] so the rooms were meant for smaller beds, smaller people, smaller equipment or less equipment. . . it's hard to walk from the bed to like say the bathroom because there's a chair in the way or there's a table in the way and forget about trying to get from you know the bed to a wheelchair and get out the door. . . it's impossible and you basically have to maneuver everything every time you go to do care. (Lavender, Healthcare provider, Site A)*

In Site B and Site C, there were also safety concerns that created restrictions on the type of patient who could be transitioned to these environments. For example, if patients were immobile and/or had responsive behaviours that required more staff support and resources (e.g., patients with dementia) there were concerns around exit seeking behaviour or the ability to evacuate patients safely in the event of a fire. For example, Site B had no sprinkler system and in the event of a fire, patients would need to be able to walk, whether independently or with a mobility aide, for evacuation:

*Unfortunately, where we are located there are some restrictions. We don't have a sprinkler system so our patients do have to be able to walk. So that really does limit our ability to take older adults who are experiencing difficulty with mobilizing and maybe some that are a little bit more acutely ill. (Chicago, Healthcare provider, Site B)*

**Table 2.** Themes with descriptions from observations, interviews example interview quotes.

| Themes | Site | Observation Notes/ Summary | Interview Notes/ Summary | Example Quotes from Interviews[i] |
|---|---|---|---|---|
| Implications on the physical safety of patients | Site A—decommissioned hospital | • Hallways were fairly clear, with some chairs, garbage bins, etc.<br>• Some bathrooms were small and not able to fit a wheelchair | • Equipment crowded hallways and could create potential injury risks for patients<br>• Inadequate space in patient rooms and on unit to provide patient care | • . . . there's equipment in places that we have to put equipment in and every day you'll see like the fire hose thing cupboard blocked because there's nowhere to put things. (Lavender, HCP)<br>• I understand that the wheelchair can't get into the bathroom. It won't go through the door (Ohio, CG) |
| | Site B–renovated hotel | • Inadequate space to provide patient care/ support<br>• Small patient rooms, with limited space for equipment<br>• Elevators too small for full-size stretcher | • No sprinkler systems<br>• Inadequate space in patient rooms and on unit to provide patient care | • . . . we'd done a lot of complaining about the bathroom because we'd go into the bathroom, we couldn't go in with the walker, we had to leave the walker out in the hall (Dean, PT)<br>• The rooms are very small. So it makes it challenging when patients have mobility aids. (Brooklyn, HCP) |
| | Site C–structured tent | • Not applicable–no observations were conducted in the tent | • Injury risk for patients (e.g., tripping on uneven surfaces)<br>• Open concept space created an opportunity for staff to provide emergency care to patients who were not on their caseload | • But even the bathrooms were like a very good deal, because we could actually wheel people on commode chairs to the bathrooms. . . And then it just gives you that sense of privacy again. You feel like you're human again. And then same thing—we can wheel the chairs right into the shower. (Jordan, HCP)<br>• We didn't have falls. I think we only had one, right. But it was somebody mobilizing around and they fell, right, it wasn't the falling out of bed, right. Because we're all right there. (Lincoln, HCP) |
| Implications on staffing models and continuity of care | Site A—old hospital | • External providers/ private physiotherapists worked with some patients<br>• Brief physician presence on floor | • Shared staffing model[ii] across three floors (improved staff coverage, but negatively impacted relationships)<br>• Limited onsite access to certain specialist physicians (e.g., psychologists) | • So, [it is] built it in as part of our culture here. Even in the interview [we] tell staff: you will work on all three floors. You will be reassigned. You could be reassigned even at the start of your shift. (Leila, HCP)<br>• . . . we don't seem to know who the doctor is and like who the attending physician is (Maple, CG) |
| | Site B–renovated hotel | • No physicians with geriatric specialties, no overnight physician coverage<br>• Knowledgeable team that can fill-in/ cover for each other | • Smaller, dedicated team improved relational continuity of care and enhanced patient experiences | • It's like just the whole team effort, everyone wants to help each other. And I just–yeah. You don't see that as often anymore because everyone's just swamped and I just feel like our patient's ratio I think is four patients per nurse—so that's fantastic because the nurses are able to go above and beyond with their patient. (Kenya, HCP)<br>• I'm lucky that I have a very supportive team. And our patients on our unit we get them up and dressed, and mobilized, as much as we can. So, for me, the patients are very accessible. So I'm able to get in in the morning, look at my list of patients, I go and meet with them. (Brooklyn, HCP) |
| | Site C–structured tent | • Not applicable–no observations were conducted in the tent | • Consistent staffing model improved relational continuity of care<br>• Assigned physician improved accessibility and integration with team | • Because it was a very open environment, the staff were right there with the patients, and it became a very social kind of place, right. And so it was really positive experience for our patients. (Lincoln, HCP)<br>• But with us we actually had assigned physician for the [unit] so it was because, you know, they had the [delayed discharge] physician group and they were kind of there. And that made things a lot easier because, you know, they are able attend rounds. They are able to be more involved with the team. (Georgia, HCP) |

(Continued)

**Table 2.** (Continued)

| Themes | Site | Observation Notes/ Summary | Interview Notes/ Summary | Example Quotes from Interviews[i] |
|---|---|---|---|---|
| Implications on team interactions and patient care | Site A—old hospital | • Frequent, hybrid rounds (face-to-face and teleconference), held in the dining room<br>• One floor had a room/lounge with colouring books, puzzles, and mental aerobics exercises for patient use; it was also used for an exercise class 1x/week | • Frequent, face-to-face team huddles (meetings)<br>• Positive interactions between patients and staff (e.g., consistent communication) | • *The nurses that I did meet that were very, very nice, really had a positive attitude. A real positive attitude because at first he didn't want to get out of bed and they kept encouraging him. . . they were very encouraging to him. (Aberfoyle, CG)*<br>• *I would say most of them [patients and caregivers] are very, very happy with the team and the communication and that sort of thing (Candice, HCP)* |
| | Site B–renovated hotel | • Face-to-face huddles/ patient rounds in the nursing station room<br>• Patients were in their rooms because there were no common areas for them to congregate<br>• Cohesive team with good morale (e.g., shared goals, consistent communication) | • Limited space for face-to-face team huddles/ rounds<br>• Formal joint assessments improved team interactions<br>• Team got to shape how unit would function<br>• Lack of communal spaces | • *I think our biggest issue at the moment is rounds. So when we have our interdisciplinary meeting it's really hard to–we don't really have a spot for it at the moment. . . we have such a small space and a small nursing station that we just don't have the room to hold the whole team (Kenya, HCP)*<br>• *I think it's just the small and intimate team we have over here. And the 20 patients you really get to know 20 patients as opposed to having a unit with 60 patients, it's really easy to know the ins and outs it's like I review the patients on a daily basis. (Dallas, HCP)* |
| | Site C–structured tent | • Not applicable–no observations were conducted in the tent | • Frequent, face-to-face team huddles<br>• Team building sessions prior to opening the unit<br>• Team got to shape how unit would function<br>• Lack of communal spaces | • *. . . we had really good rounds especially, you know, since we had a smaller number of patient population, so we were able to really, you know, have really good communication every morning. And the entire team was present for that (Georgia, HCP)*<br>• *We did quite a few orientation days and teambuilding days, and then leading up to the [unit] opening, it was seamless (Ireland, HCP)* |

[i] HCP—Healthcare provider (includes care providers, allied health professionals, rehabilitation therapists, hospital managers and staff); PT—patient; CG—caregiver

[ii] Shared staffing model: the policies, procedures, and practices used to manage staffing levels and types of staff were used across multiple floors to ensure a larger pool of staff

Although patient eligibility was restricted in some cases, patients, particularly those with mobility aids, were still faced with physical safety challenges due to the small and cluttered spaces. For example, one healthcare provider identified how it was difficult for anyone to provide patient support while accessing the bathroom:

> *The bathroom itself is extremely tiny, you can't really take gait-aides in there, so almost every single patient here, like more than 90 percent use gait-aids. So it's nearly impossible to get a gait-aid in there, so that's a problem. . . So another one, imagine like a bigger person trying to get into a tiny washroom with their gait-aid and a person to help them. (Hampton, Healthcare provider, Site B)*

In another situation, there were safety concerns for patients using wheelchairs in Site C because the environment consisted of narrow aisle ways and was built on top of the uneven ground of the hospital parking lot:

> *. . .Our fear was what if this wheelchair gets jammed, now this person is going to be falling forward. So we were very safe or safety oriented. We knew where the issues were and it's not*

*smooth. It's not like the hospital where it's nice and smooth concrete. It's upside down. And it's been salted. And then there's cracks. (Jordan, Healthcare provider, Site C)*

Even though several safety concerns were described by participants in the interviews and seen in the observations, benefits to patient safety in the unconventional environments were also noted by participants. For example, Site C had an open concept space, which allowed for providers to get to know and keep an eye on all patients, even if they were not on their case-load. As exemplified below, this open concept environment was perceived to allow active response to a patient's acute needs, who had diabetes and rapidly fluctuating blood glucose levels:

*. . . And because of the way [Site C] was set up. . . all of the patients were visible at all times. And this lady preferred to sleep in her wheelchair during the day. So, she preferred to spend her time in the wheelchair, and she would sleep kind of leaning over the side and you know, we all knew that's what she did. But one day one of the nurses, not even her nurse I don't think it was even her nurse, heard snoring and thought she doesn't usually snore and went over to check on her. If she had been in a regular unit, she would have died because nobody would have seen her, nobody would have heard her snoring, nobody would have seen and gone in and go oh, maybe she doesn't look quite right. She would have died. But because of the physical set up that was an amazing–and that was a skilled nursing assessment to pick that up and act on it. (Adele, Healthcare provider, Site C)*

## Implications on staffing models and continuity of care

The organization of staff on the units seemed to impact patient care and experiences. More specifically, the staffing models (policies, procedures, and practices used to manage staffing levels and types of staff) seemed to impact the ongoing relationship between patients and their care providers [29]. For instance, we learned from patient and caregiver participants that having a smaller, dedicated team improved relational continuity of care, better patient care, and enhanced patient experiences because "everyone has got little tidbits of knowledge" (Observation, Site B). A provider compared the uniqueness of the unconventional environment to a regular hospital unit, explaining how the consistent staffing model was beneficial in allowing the staff to get to know all patients so they could work through care challenges together. This mentality was also something they carried on when they moved onto other units:

*Because at [Site C] we had like everybody knew everything about this patient because as charge nurses, like I said, we had designated charge nurses only charge for days and nights. So we really knew everything about the patient. And we actually had a little summary on every patient that was there. But in order like to—so I don't think that's feasible on the floors just because everything is always dynamic and changing on the floors, especially on the medical units [compared to unconventional environments]. (Jordan, Healthcare provider, Site C)*

Site A originally opened on one floor; however, it expanded to an additional two floors and had a shared staffing model. The staffing across the three floors was described as one large team dedicated to all three floors. A healthcare provider discussed the perceived benefit of this model in providing coverage if one floor was short-staffed:

*I again looked [at] the three floors together as one unit, one team. So, I did develop a shared staffing model. Basically, what that means is that all of my staff here at [Site A], they're all*

*one big staffing pool. So, we do not have siloed floors. . . we have one large staffing pool, and these are the staff for [Site A] and all of my staff know and they are required to work on any of the 3 floors, depending on where we need them. . . The reason why I developed this type of staffing model was to optimize the staffing resources for all 3 floors. I knew that if we kept each floor as an individual unit we would run into significant challenges with staffing. (Leila, Manager, Site A)*

Despite the perceived benefit by staff of this model in ensuring adequate coverage across the three floors, patients and caregivers perceived a lack of continuity in care and described having a negative experience:

*And that's the other thing that makes me crazy. You have a nurse, you build up a rapport with that nurse over maybe two or possibly three shifts, then you may not see them again for a month or more, if at all. Because they have to staff all three floors, and I think that's wrong. . . Like there's a really nice male nurse and he and I built up a rapport, but I haven't seen him in probably three weeks, at least. Because he's working I think on the first floor. So I find that challenging. . . (Clark, Patient, Site A)*

Our interviews and observations also showed that having limited onsite access to physician specialists during rounds (e.g., geriatrician, psychologist) was a contributing factor to disrupting the continuity of care that patients received. As described by a healthcare provider, patients who needed access to these specialties had to be transferred to the main hospital site and seen in the emergency department. The disruptiveness of not having access to specialists onsite was highlighted by a healthcare provider, who stated:

*We don't have good access to a lot of specialties. If we want a psychiatrist to see a patient, we often will send the patient back to main site, to Emergency and they will get seen there [. . .] So again, talking about a satellite site that's quite a distance and you're bringing these very highly complex patients with behaviours out here–this is a gap for sure. (Leila, Manager, Site A)*

On the other hand, Site C was described as having timely access to a physician, as they were designated to the site and on the unit. At this site, there was better integration with the care team and patient questions were addressed quickly. Having an assigned physician was helpful for both staff and patients:

*But with us, we actually had assigned physician for the [Site C] [. . .] And that made things a lot easier because, you know, they are able [to] attend rounds. They are able to be more involved with the team. They are there if you have a quick question that you want to run by them, or if you need a form completed or signed or whatever, they're right there [. . .] so I think from the patient point of view I think probably would have made things a lot smoother like if somebody was concerned about something, we're able to get the physician to address it right away because they're right there. (Georgia, Healthcare provider, Site C)*

## Implications on team interactions and patient care

Participant interviews and observations suggested there were more opportunities for collaboration when the whole team was working together within the same environment on a regular basis. Team members were able to have more frequent informal face-to-face conversations as well as formal team huddles (although space was a limiting factor for some environments).

These increased interactions allowed team members to have more in-depth conversations about patients, provide health status updates, and discuss any changes to patient care plans. For example, a participant shared how as a team they were able to assess patients more holistically to help with transition to the next point of care and/or avoid hospital re-admissions:

*The joint assessments are working really well, and we've kind of approached it as a comprehensive geriatric assessment. So we don't just look at like–you know traditionally for physios in hospitals, it's like how is their mobility? How is their balance? And like you don't really look too, too much deeper. But we're looking at things like meds, moods, nutrition, sleep. Their ADLs [activities of daily living], their IADLs [instrumental activities of daily living]. How much homecare they receive, their memory, their cognition; so it's a lot deeper than a typical assessment. (Hampton, Healthcare provider, Site B)*

Opportunities for collaboration allowed team members to shape how the space would be used (or designed in case of Site C) to provide care in the unconventional environment. Participants described how there was time for planning and/or team building prior to the opening of these unconventional environments to help the team orient to the space and their colleagues:

*It's really nice to be a part of something right from the ground up. I've been able to be a part of the planning for this unit, be a part of, you know, having my input and my opinion asked about how can we create this amazing geriatric unit. (Chicago, Healthcare provider, Site B)*

Participants also described how everyone on the team a shared goal of providing quality care despite being in an unconventional environment. Team members also had an understanding of the reasons why patients had to be moved to these locations (e.g., relieving hospital capacity pressures due to COVID) and collaborated to make the unconventional environment work as efficiently as possible. Participants discussed how the team was committed to the care they were providing:

*The team was incredible. Yeah, we had a really, really good strong team and everybody worked really, really well together. And we were really committed to, you know, making sure that very unconventional space worked as well as any of our conventional spaces, if not better. (Georgia, Healthcare provider, Site C)*

Although team interactions may have been enhanced due to working together in close proximity, we observed that some of the unconventional environments lacked communal areas for patients to socialize with their caregivers, other patients and/or staff on the floor. As well, there were limited areas, if any, for staff to provide physical and occupational support. One participant described how they will eventually move into a new space, but in the meantime, they lacked a gym to provide physiotherapy to patients:

*We're so restricted, we don't have a gym, we don't have anywhere where, really, they can go right now [laughs]. I think when we have the actual centre where–and the rooms are tiny and the bathrooms are tiny here [laughs]. So, once they have room to move around and we can take, like I said, those patients that just need some help for a couple of days to get to that strength that they can stand appropriate. (Erie, Healthcare provider, Site B)*

Despite the challenges with limited areas for rehabilitation activities and team meetings in Sites B and C, we noticed during the onsite observations that the hallways at Site A were wide

enough to easily accommodate mobility aids and assistive devices including stretchers, wheel chairs, and walkers. At this site, the presence of hallway space to accommodate equipment and furniture allowed for certain aspects of patient care to occur (e.g., physical or occupational therapy, educational activities) and social activities (e.g., eating lunch with other patients, enjoying group entertainment), compared to sites where staff and patients were more limited by small patient rooms and lack of communal areas. When communal spaces were available, patients were also able to engage with and be part of the daily activities and flow of the unit.

## Discussion

In this multi-method qualitative study, we used interviews and observations to describe the characteristics, and explore individuals' experiences, of three unconventional environments across two geographically diverse (northern and southern) health regions for patients with delayed discharge from hospital. We identified three main interrelated themes–(1) implications on the physical safety of patients; (2) implications on staffing models and continuity of care; and, (3) implications on team interactions and patient care. Based on these findings, we have identified the following considerations when providing care to patients with delayed discharge in unconventional environments: maximizing patient safety, having communal spaces for physical activities and socialization, co-locating team members, and having an unconventional environment in close vicinity to the main hospital.

Our interviews highlighted several perceived safety concerns in these unconventional environments that could potentially increase patient harm. The environments were described as poorly designed for patient care (in the cases where the space was repurposed); having small rooms with narrow hallways that were unable to accommodate interactions between patients, caregivers, and staff; lacking proper storage for necessary equipment and medical devices; and/or lacking communal spaces for social interactions and group therapy. Given the increased isolation experienced during the pandemic [30, 31], which can contribute to deconditioning, it is possible that unconventional environments further compounded these issues. Issues with patient safety and harm in hospitals are already well documented in conventional environments [32–34], requiring immediate action. These safety concerns put patients at increased risk (e.g., deconditioning, injury, social inactivation, mortality) and can negatively impact the cost of care, discharge processes, length of stay, and patient and staff satisfaction, causing further distress to patients, caregivers, and healthcare providers [35–37].

Designing hospital environments for both safety and quality can improve patient outcomes and experiences [38–41]. Our interviews showed when the team at site C had the opportunity to contribute to the design of the environment, they were able to be proactive about potential safety issues (i.e., designing wide bathrooms/showers, open concept space to have more eyes on patients and identify safety risks). Proactively addressing patient safety aligns with the core tenets of the Measurement and Monitoring Safety Framework (MMSF) [42, 43]. The MMSF encourages healthcare teams to move away from focusing on past harm and mitigating risks to creating a culture where safety is proactively managed by staff, patients and caregivers working together [43]. Involving all stakeholders in the early planning and design of unconventional environments is important to promote a culture of safety and mitigate potential harms, while ensuring that patients, as well as their families and care team, feel confident that high quality care will still be provided.

In order to address safety concerns and create an optimal unconventional environment, constructs from the Optimal Healing Environment (OHE) framework can also be drawn upon [44]. The authors of this framework define healing as "a holistic, transformative process of repair and recovery in mind, body, and spirit" (p. 40). The concept of healing is particularly relevant for individuals who may be at risk for deconditioning and further hospital harms

when experiencing a delayed discharge [4–6], as they no longer require acute care yet typically have ongoing chronic care needs and emotional support needs requiring attention.

One component of the OHE framework focuses on the interpersonal environment, including personal and professional relationships [44]. The development of these relationships is based on open communication, awareness of power imbalances, emotional self-management, and presence during encounters. In our study, patients and caregivers noted that rotating staffing models had an impact on their ability to develop relationships with certain healthcare providers, further impacting relational continuity of care [29] and overall experiences. Similar findings regarding consistency of healthcare providers and relational continuity of care have been noted [45, 46]. For example, higher patient satisfaction has been associated with having a consistent healthcare provider [45], along with an improved ability to develop an ongoing relationship, and increased confidence in the quality of care received [46]. Further to this, the OHE framework emphasizes the promotion of wellbeing for patients and healthcare providers through person-centred approaches, developing relationships, and supporting shared decision-making [44]. As such, the creation or repurposing of a space as an unconventional environment for patient care should consider how these processes (developing relationships, delivering person-centred care, instilling shared decision-making) can be better supported through adequate resources and improved continuity of care with consistent staff.

Another component of the OHE framework focuses on the behavioural environment, which entails actions taken by oneself or others to promote healing [44]. Integrated care is a key construct of the behavioural environment, which is conceptualized as "team-based care that is person-focused and family-centered and incorporates multidisciplinary care providers at their highest skill level" (p. 42) [44]. In our study, the physical proximity of team members had positive impacts on interactions and experiences. Whereas, not having access to onsite physicians and specialists was identified as a challenge. The co-location of team members in a healthcare setting has been noted to facilitate collaboration [47, 48]. More specifically, a systematic review of multidisciplinary collaboration found that teams that were co-located in the same space were highly coordinated, leveraging meetings and face-to-face communication [47]. Similarly, a qualitative study highlighted a number of benefits of the co-location of interdisciplinary teams, including: ease of access, relationship development, team congeniality, collective efficacy, idea exchange, seamless person-centred care, and collaboration [48]. Ultimately, given the multidisciplinary nature of care teams within these unconventional environments, it is important to consider co-locating the healthcare providers on the team to ensure maximum benefit can be achieved for both patients and providers.

## Strengths, limitations and future research

This study was limited to three unconventional environments across two hospital networks and may not be representative of how these environments are structured in other hospitals. These environments were also operating ad hoc at times during the pandemic which could have affected what we observed and heard throughout interviews over the course of data collection. However, we identified very little published information about these types of unconventional environments, limiting our understanding of how they are structured and the experiences of those who are receiving or providing care within these environments. These unconventional environments are becoming increasingly common as hospitals continue to deal with capacity pressures. Our study provides important learnings about how to optimize these environments to ensure quality patient care. Future research is needed to better understand the healthcare utilization, health outcomes (including adverse events), and experiences of patients with delayed discharge in these environments.

Due to ongoing COVID-19 pandemic and visitor restrictions that were in place, interviews were conducted over Zoom or telephone, potentially limiting the observation of non-verbal cues. However, using a mix of video or phone does not undermine the quality of data collected [49–51]. We were unable to complete participant observations or conduct interviews with patients and caregivers at Site C. There were also recruitment challenges across the other two sites because we were unable to be on site frequently to interact directly with patients and caregivers about the study (i.e., during outbreaks) and instead, relied on building relationships with staff to share our study information with patients and caregivers. As a result, we do not have as many patient and caregiver perspectives as initially projected. This presents an important area of future work in further exploring and understanding patient and caregiver perspectives of receiving care in unconventional spaces. Despite the challenges, we were able to interview participants in different roles (i.e., nurses, physical and occupational therapists, social workers, discharge planners, managers, physicians, patients and caregivers), which allowed us to hear multiple perspectives about how care was being received or provided across each environment. The combination of using data from interviews and naturalistic observations was another strength of this study, as the observations helped contextualize the interview data that was collected [52], and provided visual context about how each environment was structured and the types of interactions between staff and patients and caregivers.

## Conclusions

Delayed discharges are an ongoing problem occurring across Canada and other health systems. This study explored three unconventional environments, places where patients with a delay in discharge are commonly placed, to better understand their characteristics and individuals' experiences of each environment. While unconventional environments are not optimally designed to protect patient safety, open concept spaces with smaller consistent teams, including physician presence, can potentially mitigate these risks. As unconventional environments are becoming more common, there needs to be careful consideration about how the spaces are designed and utilized to ensure patient safety and high quality of care.

## Acknowledgments

We would like to thank all the participants for generously sharing their time, experiences and knowledge to support this work. We would also like to thank Harprit Singh for her contributions and support with this project. Kerry Kuluski holds the Dr. Mathias Gysler Research Chair in Patient and Family Centred Care. Sara Guilcher holds a salary award as Pain Scientist from the University of Toronto Centre for the Study of Pain.

## Author Contributions

**Conceptualization:** Walter Wodchis, Sara J. T. Guilcher, Kerry Kuluski.

**Data curation:** Michelle Marcinow, Lauren Cadel, Jane Sandercock.

**Formal analysis:** Michelle Marcinow, Lauren Cadel, Arija Birze, Junhee Baek, Sara J. T. Guilcher, Kerry Kuluski.

**Funding acquisition:** Walter Wodchis, Sara J. T. Guilcher, Kerry Kuluski.

**Investigation:** Lauren Cadel, Sara J. T. Guilcher, Kerry Kuluski.

**Methodology:** Michelle Marcinow, Walter Wodchis, Sara J. T. Guilcher, Kerry Kuluski.

**Project administration:** Michelle Marcinow, Lauren Cadel, Jane Sandercock, Kerry Kuluski.

**Resources:** Kerry Kuluski.

**Supervision:** Sara J. T. Guilcher, Kerry Kuluski.

**Validation:** Sara J. T. Guilcher, Kerry Kuluski.

**Visualization:** Sara J. T. Guilcher, Kerry Kuluski.

**Writing – original draft:** Michelle Marcinow, Lauren Cadel, Arija Birze.

**Writing – review & editing:** Michelle Marcinow, Lauren Cadel, Arija Birze, Jane Sandercock, Junhee Baek, Walter Wodchis, Sara J. T. Guilcher, Kerry Kuluski.

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
