## [Decision Letter · Decision Letter 0]

23 Oct 2023

PONE-D-23-23636"I think we did the best that we could in the space:” A qualitative study exploring individuals’ experiences with three unconventional environments for patients with a delayed hospital dischargePLOS ONE

Dear Dr. Cadel,

Thank you for submitting your manuscript to PLOS ONE. After careful consideration, we feel that it has merit but does not fully meet PLOS ONE’s publication criteria as it currently stands. Therefore, we invite you to submit a revised version of the manuscript that addresses the points raised during the review process.

We look forward to receiving your revised manuscript.

Kind regards,

Robbert Huijsman, PhD

Academic Editor

PLOS ONE

Additional Editor Comments:

We thank the authors for their paper about unconventional environments for patients with a delayed hospital discharge. Reviewer 1 has done excellent work to provide you with comments and suggestions. Additionally, the Academic Edtior has some additional remarks. Firstly, the covid-crisis is the background context which initiated indeed such a unconventional environment like a heated tent, but an other environment like a hotel is not that unconventional in normal non-covid times. Your paper may gain strength by reflecting on the pros and cons of these various kinds of environments to reduce hospital stay in normal times; the three themes seem to be generalizable, by embedding it more in literature. Secondly, your Discussion is a bit long and wordy, please try to make it more condense. Lastly, your quotes form interviews are rather long, please shorten them where ever possible.

Reviewers' comments:

Reviewer's Responses to Questions

**Comments to the Author**

1. Is the manuscript technically sound, and do the data support the conclusions?

Reviewer #1: Partly

2. Has the statistical analysis been performed appropriately and rigorously? 

Reviewer #1: N/A

3. Have the authors made all data underlying the findings in their manuscript fully available?

Reviewer #1: Yes

4. Is the manuscript presented in an intelligible fashion and written in standard English?

Reviewer #1: Yes

5. Review Comments to the Author

Reviewer #1: The authors demonstrated good effort in the conduct of this study by using a combination of observation and interview techniques to confirm the findings. However, the writing skills needed improvement, especially in the flow and coherency. The skill in the presentation of the results needed improvement as well. The current results are considered ''first draft''. More literature search is needed to improve the ''theme'' and related result presentations.

6. PLOS authors have the option to publish the peer review history of their article (what does this mean?). If published, this will include your full peer review and any attached files.

Reviewer #1: No

---

## [Author Response · Author response to Decision Letter 0]

4 Jan 2024

Editor Comments:

Firstly, the covid-crisis is the background context which initiated indeed such a unconventional environment like a heated tent, but an other environment like a hotel is not that unconventional in normal non-covid times. Your paper may gain strength by reflecting on the pros and cons of these various kinds of environments to reduce hospital stay in normal times; the three themes seem to be generalizable, by embedding it more in literature.

Response: We have revised the introduction and added detail regarding the use of unconventional environments prior to the pandemic. Unconventional environments are indeed becoming more common and despite being used during non-COVID times, these sites are understudied and will continue to be used in the future. Therefore, it is important that we better understand individuals’ experiences with them.

Secondly, your Discussion is a bit long and wordy, please try to make it more condense.

Response: We have revised the discussion to make it more concise and less wordy.

Lastly, your quotes form interviews are rather long, please shorten them where ever possible. 

Response: We have shortened the quotes in the results.

Reviewer 1 Comments

Overall comment

The authors demonstrated good effort in the conduct of this study by using a combination of observation and interview techniques to confirm the findings. However, the writing skills needed improvement, especially in the flow and coherency. The skill in the presentation of the results needed improvement as well. The current results are considered ''first draft''. More literature search is needed to improve the ''theme'' and related result presentations. 

Response: Thank you for these comments. We have revised the manuscript to improve the writing (flow and coherency), as well as the presentation of the results.

Introduction: 

•The introduction did not provide an in-depth and clear perspective that the unconventional environments were only referring to pandemic periods. 

Response: We have revised the introduction to improve clarity that the perspectives of the unconventional environments were referring to the pandemic periods

•Secondly, the flow of the introduction could be further improved. 

Response: Thank you for this comment. We have reorganized the introduction to improve flow.

•Thirdly, the authors need to describe the context of delayed discharge. The readers in other countries may have different interpretations of the meaning of delayed discharge. 

Response: We have added more context to better describe delayed discharge to ensure readers from different countries interpret the meaning in the same way. 

•Also, the structure and under what conditions the delayed discharge patients need to be admitted to unconventional environments.

Response: We have added detail to address the conditions under which patients with delayed discharge were admitted to unconventional environments.

The first paragraph gave the reader an impression that the context of the manuscript was focusing on general perspectives of delayed discharge. The context of delaying hospital discharge should focus on the pandemic period in the first paragraph. For example, a sentence in the third paragraph, ''Patients experiencing a delayed care transition, also known as a delayed discharge or alternate level of care, are a population that was frequently transitioned to unconventional environments during the pandemic,'' should be brought forward to the first paragraph. 

Response: Thank you for this comment. We have reorganized the introduction to improve readability. 

Objective

The objectives were unclear; the methods were mainly focused on the experiences during the pandemic, but it was not specified in the objectives per se. Please improve. 

Response: The objectives were outlined as follows: (1) describe the characteristics of three unconventional environments and (2) explore individuals’ experiences with the three unconventional environments.

While data collection occurred during the pandemic, the objective was not to focus on experiences specific to COVID-19. Please refer to the second objective – to explore individuals experiences with the unconventional environments. Unconventional environments were used prior to the pandemic but have become increasingly common in recent years. Additionally, since the study took place during the pandemic, it may be presumed that this event shaped the experiential data.

''describe the characteristics of three unconventional environments'': there were no prior sentences mentioning three unconventional environments. It is, therefore, uncertain which three unconventional environments were referring to. Response: Thank you for noting this. We have added the three environments in brackets, as the details of them are expanded on in the methods.

Methods

''Unit level observations'': 

•How was observation being performed at site C? 

Response: Observations were not conducted at Site C because of visitor restrictions, the site would not allow researchers to enter. We have added detail to clarify this.

•Were the patients or their caregivers at the sites aware of the presence of the study team members during the conduct of the observations? 

Response: Patients and caregivers were informed that study members were onsite during the observations; we have added this detail.

•Video conferencing: were the videos being turned on during the interview? 

Response: The interviewer always had their camera turned on during the interview, but participants were not required to. These details have been added.

Interview

•Were the providers and clinical managers recruited being paired? Were the patients and caregivers recruited being paired?

Response: Recruited participants were not being paired (e.g., providers were not paired with their clinical manager and patients were not being paired with their caregiver). There was opportunity for paired individuals to participate, but this was not a requirement. We have added this clarification.

•''participant experiences with receiving or providing care'': the author may need to include the topics, dimensions, or examples of questions covered in the interview. 

Response: We have added additional details around topics explored in the interviews.

Data analysis

•Although it was being cited, it would be better to include more information about ''leading practices guide for delayed discharge'' including providing the Title of the guide.

Response: We have added additional detail about the leading practices guide, including the title.

•''model'': The term ''model'' appeared abruptly in the data analysis without prior introduction. Please improve. The term model of care'' should be brought up in the introduction. Lack of coherency when ''model'' appeared abruptly in the data analysis section.

Response: Thank you for noting this – we have revised the introduction to include and define this term (model).

•''In combination with publicly available documents'': what do the publicly available documents refer to? The author may want to clarify this.

Response: We have added examples to improve clarity around what was meant by publicly available documents. 

•''Observations were made during rounds, meetings, huddles, and overall day-to-day functioning.'' this sentence should be included in the method of data collection. 

Response: Thank you for noting this, we have moved this sentence in the methods.

•''Further codes were used in order to inform the descriptions outlined in Table 1 of the results: context of the model of care, physical space, staff composition, patient eligibility, and participant role. The information gathered from these codes helped to inform the descriptions outlined in Table 1 of the results.'' Comment: The flow was disrupted. It appeared unclear to the reviewer with regard to what ''descriptions'' were outlined in Table 1. What does ''further codes' refer to? The reviewer cannot relate the meaning of further and the description outline in Table 1. It wasn't very clear. Normally, when mentioning Table 1 in the text, ''Table 1'' should be followed at the end of this paragraph. The author may need to improve the flow. 

Response: Thank you for bringing this to our attention. We have revised this section to remove reference to table 1 to avoid disruption and improve flow.

Saturation points: how were the saturation points being determined? 

Response: We have added this detail to the methods.

Credibility and trustworthiness: please describe how these two aspects are being ensured in the process of data analysis. Response: Thank you for noting this. We have added this information to the methods section. 

Results

Overall comments about results: The descriptions are fair; they can be improved in order to make the overall presentation succinct. The definition or description of the meaning of each theme that is derived from the ''leading practice guide'' can be added at the beginning of each theme description. 

Response: Thank you for this comment. We would like to clarify that the leading practices guide was used deductively to inform the coding process, but it is not linked to the final themes that are presented in this paper. As such, we are unable to add a definition and/or description of the themes related to the guide (as the themes are not connected to the leading practices guide).

''A total of 29 individuals participated in this study from the three unconventional environments. Comment: please improve this sentence. 

Response: We have revised the sentence. 

Table 1

•The physical environment column can be improved. The current presentation is disorganized and messy. 

Response: We have revised the physical environment column of Table 1 to improve organization and presentation of information.

•Site A: ''Older hospital, renovated in 2018 to address capacity pressures.'' This information did not help the reader to understand the context of A in a better picture. 

Response: We have revised the description of Site A to improve understanding for the reader.

Table 2

•The interview notes/ summary column needed improvement (more data ''cleaning'') for concise information. 

Response: Thank you for noting this. We have revised the interview notes/summary column to clean it up and make more concise. 

•Staffing model: what is the ''staff model''?

Response: We have added the definition of a shared staffing model.

•Frequent, face-to-face team huddles (by floor): what does this mean? 

Response: We have removed the reference to “by floor” to eliminate confusion. Huddles are a term for team meetings and we have added this detail.

Theme: Implications on staffing models and continuity of care

•Comment: kindly address the definition of ''staffing model'' at the beginning paragraph of this theme.

Response: We have added the definition of a staffing model.

•What is the meaning of the ''shared staffing model.''?

Response: We have added the meaning of a shared staffing model. 

•The last paragraph of the third theme: suggest adding a quote for the description of the last paragraph. 

Response: While we appreciate this suggestion to add a quote for the last paragraph of this theme, this description is based on the observations, so we do not have verbatim quote to provide.

Discussion:

•What do you mean by ''two diverse health regions''?

Response: We have added context to clarify what is meant by diverse health regions.

•Page 26-27, lines 465-472: The author may want to revise this paragraph. The ''healing'' context is important, but the description lacks linkage to the context of the unconventional environment; the authors only linked ''healing'' with delayed discharge. 

Response: We have revised this section to emphasize the link between healing and delayed discharge. We have also removed the paragraph specific to healing based on this comment and a recommendation from the editor to shorten the discussion.

•''One component of the framework focuses on the interpersonal environment, including personal and professional relationships.'' Which framework was the author referring to?

''Another component of the framework focuses on the behavioural environment, which entails actions taken by oneself or others to promote healing.'' Again, which framework was it referring to? 

Response: These references were in relation to the Optimal Health Environment Framework introduced in the prior paragraph. We have added this to clarify.

•The prior sentence was followed by ''Integrative care''. There was a lack of coherency between the prior sentence and ''integrative care''. The audience cannot relate to the connectivity between these two sentences. 

Response: The connection of integrated care is to the behavioural environment, introduced in the prior sentence. We have revised to clarify this connection for readers. 

Conclusion

•''We used constructs from the OHE framework and MMSF to provide insights…'' This OHE framework and MMSF appeared abruptly in the conclusion without prior introduction of these two frameworks in the prior sections. This renders it difficult for the reader to relate the frame to study findings and conclusions. 

Response: While we introduced these frameworks in the discussion, we appreciate the abruptness of them in the conclusion and have removed this section.

---

## [Editor Report · Decision Letter 1]

9 Jan 2024

"I think we did the best that we could in the space:” A qualitative study exploring individuals’ experiences with three unconventional environments for patients with a delayed hospital discharge

PONE-D-23-23636R1

Dear Dr. Cadel,

We’re pleased to inform you that your manuscript has been judged scientifically suitable for publication and will be formally accepted for publication once it meets all outstanding technical requirements.

Kind regards,

Robbert Huijsman, PhD

Academic Editor

PLOS ONE

Additional Editor Comments (optional):

The authors have done a good job in carefully revising their paper and responding to the comments of the reviewers.

---

## [Editor Report · Acceptance letter]

17 Feb 2024

PONE-D-23-23636R1 

PLOS ONE

Dear Dr. Cadel, 

I'm pleased to inform you that your manuscript has been deemed suitable for publication in PLOS ONE. Congratulations! Your manuscript is now being handed over to our production team.

Kind regards, 

on behalf of

Professor Robbert Huijsman 

Academic Editor

PLOS ONE